# Prolonging Bacterial Viability in Biological Concrete: Coated Expanded Clay Particles

**DOI:** 10.3390/ma14112719

**Published:** 2021-05-21

**Authors:** Ronaldas Jakubovskis, Augusta Jankutė, Simona Guobužaitė, Renata Boris, Jaunius Urbonavičius

**Affiliations:** 1Institute of Building and Bridge Structures Laboratory of Innovative Building Structures, Vilnius Gediminas Technical University, 10223 Vilnius, Lithuania; augusta.jankute@vilniustech.lt (A.J.); simona.guobuzaite@vilniustech.lt (S.G.); 2Department of Chemistry and Bioengineering, Vilnius Gediminas Technical University, 10223 Vilnius, Lithuania; jaunius.urbonavicius@vilniustech.lt; 3Laboratory of Composite Materials, Institute of Building Materials, Vilnius Gediminas Technical University, 10223 Vilnius, Lithuania; renata.boris@vilniustech.lt

**Keywords:** biological concrete, self-healing concrete, bacterial viability, crack healing, expanded clay

## Abstract

One of the biggest challenges in the development of a biological self-healing concrete is to ensure the long-term viability of bacteria that are embedded in the concrete. In the present study, a coated expanded clay (EC) is investigated for its potential use as a bacterial carrier in biological concrete. Eight different materials for coatings were selected considering cost, workability and accessibility in the construction industry. Long-term (56 days) viability analysis was conducted with a final evaluation of each coating performance. Our results indicate that healing efficiency in biological concrete specimens is strongly related to viable bacteria present in the healing agent. More viable bacteria-containing specimens exhibited a higher crack closure ratio. Our data suggest that the additional coating of EC particles improves long-term bacterial viability and, consequently, provides efficient crack healing in biological concrete.

## 1. Introduction

Biological concrete combines the advantages of traditional concrete and autonomous crack repair. The self-healing effect is achieved by using calcium carbonate (CaCO_3_) precipitating bacteria. When the cracks open, oxygen and water start to penetrate the concrete, which easily triggers the metabolic activity of bacteria. Several biological pathways have been proposed for the precipitation of CaCO_3_, such as metabolic conversion of organic acids, ureolysis or denitrification [1]. As a result, cracks may be autonomously sealed in just two or three weeks, avoiding high-cost repair technologies and constant maintenance [2,3].

A biological concrete healing approach has attracted the attention of scientists and entrepreneurs in the last decade [4]. It manifests as a very promising, effective and sustainable solution for crack healing due to several important reasons: (i) the bacteria-precipitated calcium carbonate strongly adheres to the crack surface causing strength regain [5], (ii) the newly formed CaCO_3_ is fully compatible with the concrete matrix [1] and (iii) the healing of cracks up to 0.8–1 mm in width is feasible [6]. 

One of the biggest challenges in the development of a biological self-healing concrete is to prolong the viability of bacteria that are integrated into the concrete. For self-healing to take place, bacterial spores must not only survive mechanical stress during the concrete mixing and hardening but also tolerate extreme pH conditions that may reach as high as pH 13 [7,8]. Moreover, a number of metal oxides present in cement as impurities may act as bactericidal agents [9]. Therefore, the healing process is possible only if bacterial spores lay in a dormant state and reactivate themselves once the environmental stimuli trigger (i.e., the nutrients start to flow through the opened crack). Although bacterial spores are resistant to multiple agents such as high temperature, extreme freezing, ultraviolet radiation, desiccation or chemical disinfectants, as strange as it may seem, a concrete matrix is too aggressive [10,11]. As a result, bacterial spores need an extra coating to stay viable. One of the ways to protect bacterial spores from concrete surroundings is based on the encapsulation technique [5]. A functional carrier should form a protective shell, not interfere with the bacterial activity and be compatible with the concrete [12].

Expanded clay (EC) and expanded perlite (EP) have been widely investigated for their potential use as bacterial carriers [6,13,14]. These materials are low cost, light and, most importantly, able to form a strong bond with the concrete matrix [14]. In addition, both EC and EP are extremely porous. As for self-healing concrete, a distinctive porosity is highly beneficial, as it allows bacterial spores to be immobilized. It has been shown that bacterial spores penetrate and attach to the internal pores of an EC easily, demonstrating excellent survivability [15]. Moreover, EC-embedded bacteria have been proved to survive negative temperatures up to 20 °C for a 60-day period [9]. Nonetheless, once EC particles are mixed into the concrete replacing a portion of a coarse aggregate, the decrease in bacterial viability is almost 100-fold [14]. Despite the aforementioned advantages of EC and EP porosity, this sponge-like structure may also facilitate the diffusion of metal oxide nanoparticles, thereby gradually increasing the toxicity to bacteria. Our preliminary study confirmed that concrete-embedded bacterial mortality may be attributed to the presence of different metal oxides (Fe_2_O_3_, ZnO_2_ and CuO) in the cement. In fact, the actual concentration of these metal oxides may exceed the claimed minimal inhibitory concentration (MIC) several times, causing rapid viability loss [16].

It appears that protection of bacteria by immobilization into the EC or EP alone may be insufficient, as bacterial viability drastically decreases during the early setting (0–7 days) of concrete [14]. To solve this problem, an additional coating of EC or EP particles has been suggested [2,14,17,18]. Styrene–acrylic base, water glass base, urethane base and asphalt base are known as potential coatings materials for EC particles [14]. In terms of workability, viscosity, setting time and price, the styrene–acrylic emulsion is qualified as the most suitable coating [14]. It has been shown that bacterial survivability in mortar made with styrene–acrylic-coated bacteria-embedded EC particles increased by almost 100-fold compared to uncoated ones [14]. Unfortunately, protective features of other EC coatings have not been reported yet.

Similarly to EC, several different coatings for EP have been explored [17]. Coatings made of low-alkaline materials such as magnesium potassium phosphate cement or acidic sulfoaluminate cement significantly increase the healing efficiency by reducing the water permeability or even completely sealing the cracks in the concrete specimens [17]. Conversely, high-alkaline coatings (e.g., ordinary Portland cement or geopolymer) have been reported to be ineffective, as the healing is comparable to that of unprotected EP [17]. According to other papers, EP coatings composed of soluble silicate, metakaolin, styrene–acrylate emulsion and water decrease water absorption by 16-fold compared to uncoated control [18]. As stated, this coating protects the immobilized bacteria and increases the crack sealing efficiency in concrete specimens [18]. Regardless, lack of data on the impact of various protective coatings on bacterial viability prevents the coating-based encapsulation technique to evolve.

Here, we report a detailed study on the influence of an additional coating layer of bacteria-embedded EC particles on the bacterial viability in a biological concrete. To evaluate the performance of coating materials, the *B. pseudofirmus* bacterial strain, commonly used as a producer of a healing agent, was selected [8,19,20]. Considering the cost, workability and accessibility in the construction industry, eight different materials for coatings were selected and tested. Long-term (56 days) viability analysis was conducted with a final evaluation of each coating performance. According to our data, there is a direct correlation between the healing efficiency and viable bacterial cells present in the healing agent: the higher the number of viable bacteria, the more sensitive response of crack closure is.

## 2. Materials and Methods

### 2.1. Bacteria Cultivation and Spore Preparation

Alkaliphilic *Bacillus pseudofirmus* DSM 8715 was purchased from the German Collection of Microorganisms and Cell Cultures, Braunschweig, Germany. Bacteria were routinely cultured in liquid alkaline nutrient medium according to the manufacturer’s recommendations. Alkaline nutrient medium contained 5 g/L of peptone, 3 g/L of meat extract, 0.42 g/L of NaHCO_3_ and 0.53 g/L of Na_2_CO_3_. Bacteria were incubated at 30 °C overnight.

Spores were prepared in a sporulation medium containing 3.5 g/L of sucrose, 4 g/L of yeast extract, 0.02 g/L of KH_2_PO_4_, 0.166 g/L of CaCl_2_, 0.476 g/L of KCl, 0.2 g/L of MgSO_4_·7H_2_O, 0.2 g/L of MnSO_4_·H_2_O, 4.2 g/L of Na_2_CO_3_ and 5.3 g/L of NaHCO_3_. Bacteria were aerobically incubated at 30 °C with shaking until the concentration reached 10^9^ cells/mL. The sporulation of *B. pseudofirmus* was quantified using light microscopy after staining [21]. After 4 days of growth, cultures containing a high number of spores were harvested by centrifugation. A spore suspension was washed twice with sterile washing buffer (10 mM Tris-HCl buffer, pH 9). To eliminate the vegetative cells, the suspension was heated for 30 min at 80 °C and washed twice. The resulting spore suspension was serially diluted in washing buffer. Aliquots of suitable dilutions were spread on alkaline nutrient agar and grown at 30 °C overnight. The colonies were counted after 24 h.

### 2.2. Bacteria Immobilization into Expanded Clay

EC (Liapor 4–8 mm) was impregnated by vacuum pressure of 0.1 MPa. Impregnation solution contained 80 g/L of calcium lactate pentahydrate, 1 g/L of yeast extract and 1 × 10^8^ CFU/mL of bacterial spores. EC particles were dried at room temperature for 72 h until constant mass. Scanning electron microscope (SEM) (JEOL JSM-7600F, Tokyo, Japan) images of dried EC particles are shown in Figure 1. According to SEM analysis, bacterial spores of a 1 µm size range successfully penetrated the EC particles and attached to its surface (Figure 1B). Finally, 2.2 × 10^7^ bacterial spores were immobilized into 1 g of dry EC on average. 

### 2.3. The Formation of a Protective Layer

Eight different coatings listed in Table 1 were tested for the production of bacteria-immobilized EC particles. Coating No. 1 (MKPC) was composed of 45 g of water, 22.5 g of potassium dihydrogen phosphate, 7.5 g of magnesium oxide and 3 g of styrene–acrylate emulsion, as described in [17]. In general, the chemical composition of this coating is similar to that of potassium magnesium phosphate cement, used as an alternative to the traditional Portland cement [22]. Coatings No. 2 and No. 3 (A1 and A2) were made of one and two layers of styrene–acrylate emulsion (Weberfloor 4716, Saint-Gobain, Paris, France), respectively. It is a water-based emulsion used as a pore sealer on concrete surfaces. Coatings No. 4 and No. 5 (P1 and P2) were composed of one or two layers of latex–acrylic composite paint (Optiva 3 Ceramic, Tikkurila, Vanda, Finland), respectively. Coating No. 6 (MgO) was produced of a moderately reactive MgO (5 g) and water (10 g) mixture. This suspension was stirred for 10 min at room temperature, which possibly led to a partial hydration rate of MgO of about 10–30% [23]. The short exposure to water ensured that MgO constituted the largest part of coating No. 6. This, in turn, ensured that MgO further reacted with water present in the concrete, forming a protective layer of magnesium hydroxide. The two remaining coatings No. 7 and No. 8 (AM1 and AM2) were prepared by mixing the styrene–acrylate emulsion (Weberfloor 4716, Saint-Gobain, Paris, France) and MgO at a ratio of 10:1 or 30:1, respectively. Specimen No. 9 (C) was uncoated and served as control. The last specimen No.10 (CC) was used as a double-negative control and was prepared with no coating and bacteria.

To distribute the coating material evenly, EC particles were immersed into a prepared protective solution. Then, the coatings were left to air-dry for 48 h at room temperature. All studied coatings are shown in Figure 2.

### 2.4. Preparation of the Concrete Samples

Cylindrical concrete specimens (Figure 3A) were cast from local sand (0/4 mm), white CEM-I type Portland cement (Aalborg White^®^, Aalborg, Denmark), tap water and EC aggregates (4/8 mm, Liapor GmbH, Hallerndorf, Germany). Concrete mix proportion was adopted from [13] and is given in Table 2. This mix proportion was used to produce concrete samples with both coated and uncoated EC particles. To ensure that the same amount of a healing agent is present in all samples, bacteria-immobilized EC particles were weighted before the coating procedure. By doing so, the total weight of the active healing agent remained the same for all concrete samples, whereas each coating added a particular additional weight to the bacteria-immobilized EC. For each coating listed in Table 1, six concrete samples were cast. Three samples were used for viability studies and three for monitoring the crack closure.

Each concrete sample was mixed in a separate plastic cup, sealed and left to cure for 7 days (Figure 3A). After the curing period, specimens were split using a vise, as described in [24]. A radial tension crack separated the sample into two parts (Figure 3A). To keep the separated parts together, the split samples were tightened using waterproof power duct tape. To avoid a complete closure of cracks, 0.1 mm crack spacers were inserted before tightening the cracked specimens (Figure 3A). Finally, the cracked specimens were immersed in tap water (Figure 3B). The pH of the water was periodically measured during the incubation period.

### 2.5. Scanning Electron Microscope Analysis

The surface of the coated bacteria-immobilized EC particles was examined using JEOL JSM-7600F Field Emission Scanning Electron Microscope (SEM) at an accelerating voltage of 10 kV. The distance between the SEM and the specimen surface was 7–11 mm. Imaging with secondary electron detectors (SEI and LEI) was used to recreate the surface topography. Samples were completely dried in the oven at 60 ± 5 °C for 72 h. The dried samples were covered with an electrically conductive thin layer of gold by evaporating a gold electrode in vacuum using a Quorum Q150R ES instrument (Quorum, Laughton, UK). 

### 2.6. Bacterial Viability Studies

The survival rate of bacterial spores in the concrete matrix was determined in 7, 14, 21, 35 and 56 days after the preparation of concrete specimens. For each viability test, 6.53 g of concrete was crushed to powder, which contained approximately 1 g of bacteria-embedded EC. The powder was suspended in sterile 10 mM Tris-HCl buffer (pH 9) and homogenized by vortexing. Then, samples were serially diluted in the same buffer. Aliquots of suitable dilutions were plated on alkaline nutrient agar and incubated at 30 °C overnight. The colonies were counted after 24 h.

### 2.7. Measurement of Crack Healing

Crack healing progress was measured after 7, 14, 28 and 56 days of the healing period in all concrete samples. To avoid the reflective brightness from water, specimens were dried one day before measurement. Crack images were taken using a digital microscope equipped with a camera (DTX 90, Levenhuk, Tampa, FL, USA). Two or three measuring areas were selected in each sample and marked using waterproof markers. To ensure that the results were representable and correct, each crack location was selected carefully: parallel cracks, missing aggregates or pieces of cement matrix as well as other irregularities were avoided. Four measurement points were selected in each area. An example of crack measurement with a target area and measuring points is presented in Figure 3B.

## 3. Results and Discussion

### 3.1. Bacterial Viability

Our preliminary data have shown that bacterial viability drastically decreases at an early age (0–7 days) [16]. After this period, the number of viable bacterial spores stabilizes. Here, we focus on the long-term protective capability of different coatings, thus the viability tests start from Day 7. The total number of viable bacteria was estimated in each specimen 7, 14, 21, 35 and 56 days from casting.

The main results of the bacterial viability in concrete specimens with differently coated EC particles are illustrated in Figure 4. The initial number of bacterial spores was equal in all specimens (2.2 × 10^7^ CFU/g, see Section 2.2). Notably, a 10-fold or, in some cases, even 100-fold decrease in the number of viable spores was observed in the first 7 days of concrete curing (Figure 4). Similar results were reported in [14]. After the initial 7-day period, the number of viable bacterial spores stabilized. 

The potassium magnesium phosphate cement (MKPC)-based coating had almost no impact on the bacterial viability whatsoever (Figure 4A). Surprisingly, our results on the MKPC coating appear to contradict the previously published data [17]. The major technological difference is that we used the immersion technique rather than the spraying technology described in [17], which may be the underlying reason for the absence of a protective MKPC effect in our hands. Similarly, the water-based latex–acrylic-composite coatings (P1 and P2) had a minor defensive outcome on bacteria, as the number of CFU was only slightly higher than that of the control samples (Figure 4D,E). It seems likely that the shell of this composite dissolved in the wet environment, losing the protective and pore-filling properties right after EC particles were blended into fresh concrete. Interestingly, the two layers of acrylic-composite (P2) increased bacterial viability for at least the first 7–14 days (Figure 4E). Even so, from Day 14, the estimated number of CFU was similar in both cases (P1 and P2). This, in fact, suggests that additional time is required to break down the second protective layer of Coating P2.

The remaining five coatings (A1, A2, MgO, AM1 and AM2) demonstrated considerably good shielding activity, increasing the bacterial survivability by almost 10-fold (Figure 4B,C,F–H). As can be seen from Figure 2B, styrene–acrylate emulsion (A1) formed a visible glossy film on the surface of EC, sealing the pores. The thickness of this protective covering varied from 8 to 14 µm, as depicted in Figure 5B. Interestingly, it turned out that there was little to no difference whether one or two layers of styrene–acrylate emulsion was coating the bacteria-embedded EC particles, as the viability results were rather similar (Figure 4B,C). This suggests that the mechanical blocking of the surface pores of EC particles is sufficient for the protection of the bacterial spores (Figure 5A,B).

High protective characteristics were also detected for the MgO-based coating, although a visible uniform shell, as with styrene–acrylate emulsion, was not observed during the preparation of the coating (Figure 2F). As described in the Materials and Methods section (Section 2), this shell contains some unreacted magnesium oxide. When mixed with fresh concrete, magnesium oxide reacts with water, forming a stable defensive covering of magnesium hydroxide [25]. The formed hydroxide crystals that are basically insoluble could effectively seal the pores of EC preserving the bacterial spores. Moreover, as the hydration of MgO is a slow and expansive reaction [26], it may prolong the lasting protection of bacterial spores. SEM analysis of the surface of the MgO-based coating confirms the pore sealing activity (Figure 5C). Because the adhesive properties of both the styrene–acrylate emulsion and MgO-based coatings were proven, we aimed to evaluate the effect of mixing these two materials with 10:1 and 30:1 ratios (Coatings AM1 and AM2, respectively). Although these newly formulated coatings expressed substantial shielding, the bacterial survivability was similar to that of the coatings produced of either styrene–acrylate emulsion or magnesium oxide alone (Figure 4G,H). The additive effect of these two materials combined was not observed during viability studies.

Even though this viability study generated fairly scattered results, all tested coating materials may be divided into two groups: coatings displaying low protection and highly protective coatings (Figure 6, Group 1 and Group 2, respectively). The former group includes those coatings that were shown to be comparable to the control specimens: P1, P2 and MKPC. The latter group represents the coatings with a significantly higher (at least 10-fold) protection power for bacteria in comparison to the control specimens. These are A1, A2, MgO, AM1 and AM2. Summarized viability data of all tested coatings are presented in Figure 6.

Altogether, all five coatings from Group 2 ensured protection for bacteria in the harsh concrete environment, Coating A1 being a winner, with the highest number of viable bacteria (Figure 5). However, more data, including the impact of different bacterial strains, concrete mix and curing conditions, as well as sample age and size, are required to give final conclusions. In fact, the nature of viability testing and sample preparation (see Section 2.4 and Section 2.6) makes the data dependent on the amount of healing agent that is collected in 6.53 g of the concrete sample. As such, a larger amount of collected EC particles may even result in a growing number of CFU over time, as can be noticed from Figure 4 and Figure 6. Regarding technical details, one layer of styrene–acrylate emulsion (Coating A1) and magnesium oxide (coating MgO) were the easiest to prepare and process. It should be noted that magnesium oxide improves the autogenous self-healing capacity, as an expansive product of MgO may additionally fill the cracks [27,28]. It is possible that the MgO-based coating not only protects the bacteria but may also directly contribute to the crack healing process itself. This effect was microscopically studied and is discussed in the next section.

### 3.2. Crack Healing

Along with bacterial viability studies, crack healing in concrete specimens with coated EC particles was optically assessed. The healing progress was monitored at 14, 28 and 56 days after concrete samples’ preparation. The representative results are summarized in Figure 7. Healing ratio H_c_ was used to estimate the healing efficiency [29]. H_c_ may vary between 0 (no crack closure) and 1 (full crack closure). 

Crack healing progress in the double-negative control specimen is depicted in Figure 7A. The healing ratio in this sample was equal to 0.26 after 56 days of being immersed in water. This confirms that concrete may heal itself to some extent autonomously, without any active healing agents [5]. A higher healing ratio (H_c_ = 0.31) was observed after 56 days in the specimen with embedded bacteria yet no coating (Figure 7B). The number of viable cells in this sample varied between 2 × 10^5^–4 × 10^5^ CFU/g (Figure 4J). The favorable pH conditions of the aquatic environment (it varied between pH 8.5 and 9 during incubation) stimulated the bacterial activity and precipitation of CaCO_3_. This, in turn, ensured a higher healing ratio in comparison to double-negative control. Significantly higher healing ratios were observed in concrete specimens with coated EC particles. Figure 7C,D illustrates the healing progress in samples with styrene–acrylate- and MgO-based coatings, respectively. A greater number of viable bacteria in comparison to the uncoated control (see Figure 4B,F) enhanced the formation of CaCO_3_. The healing ratio of 0.84 and 0.60 was achieved in 56 days for styrene–acrylate- and MgO-based coatings, respectively. As to MgO, the formation of Mg(OH)_2_ may improve long-term autogenous healing. A more detailed long-lasting statistical analysis with a larger sample number and broader variety is required to verify this effect.

Nevertheless, our results indicate that healing efficiency in biological concrete specimens is strongly related to viable bacteria cells present in the healing agent. More viable bacteria-containing specimens exhibited a higher crack closure ratio. Consequently, additional coating of EC particles is essential to prolong bacterial viability and enhance crack healing in biological concrete.

## 4. Conclusions

In the present study, a detailed investigation on the viability of *B. pesudofirmus* bacteria in concrete specimens produced with differently coated EC particles was carried out. Optical measurement of crack healing in biological concrete specimens was also performed. Based on the aforementioned data, the following conclusions may be drawn. First, a concrete matrix is a harsh environment even for EC-embedded bacterial spores. Notably, the porous structure of EC may facilitate the diffusion of the metal oxide nanoparticles that at certain concentrations are lethal to bacteria. Moreover, a surface coating of EC particles may improve bacterial viability by almost 10-fold. To ensure the high survival rate of bacteria, the coating material must block the surface pores of EC. In addition, styrene–acrylate- and MgO-based coatings were characterized as excellent protectors. Indeed, these materials may be further used for the development of a coating-based encapsulation technique. Undoubtedly, future research is needed to assess the protective capabilities of styrene–acrylate- and MgO-based coatings for other bacterial strains suitable for biological concrete production. Finally, healing efficiency in biological concrete specimens is related to viable bacteria cells present in the healing agent. The more viable bacteria, the higher the crack closure ratio is.

## Figures and Tables

**Figure 1 materials-14-02719-f001:**
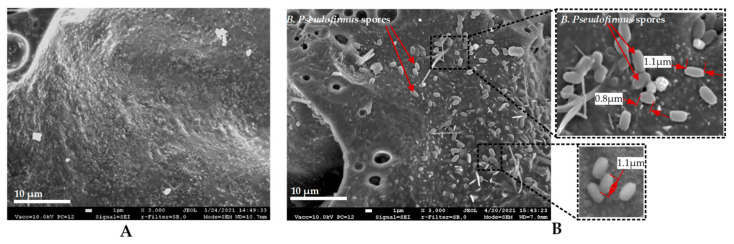
Scanning electron microscopy images of an expanded clay: (**A**) control specimen; (**B**) with immobilized *B. pseudofirmus* spores.

**Figure 2 materials-14-02719-f002:**
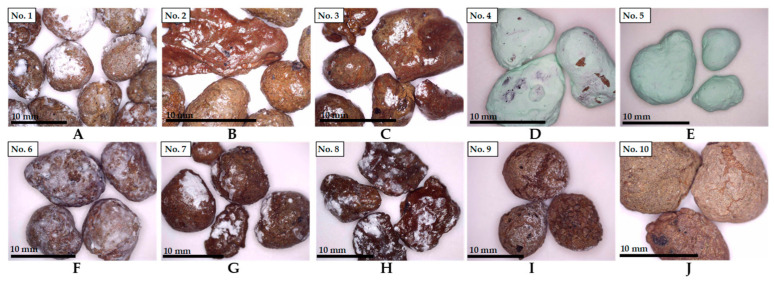
Expanded clay particles used in this study. (**A**–**H**) Particles coated with: (**A**) potassium magnesium phosphate cement; (**B**,**C**) one and two layers of styrene–acrylate emulsion, respectively; (**D**,**E**) one and two layers of waterborne, latex and acrylic-composite paint, respectively; (**F**) magnesium oxide and magnesium hydroxide; (**G**,**H**) styrene–acrylate and magnesium oxide solution, prepared with ratios 10:1 or 30:1, respectively; (**I**) uncoated control; (**J**) double-negative control.

**Figure 3 materials-14-02719-f003:**
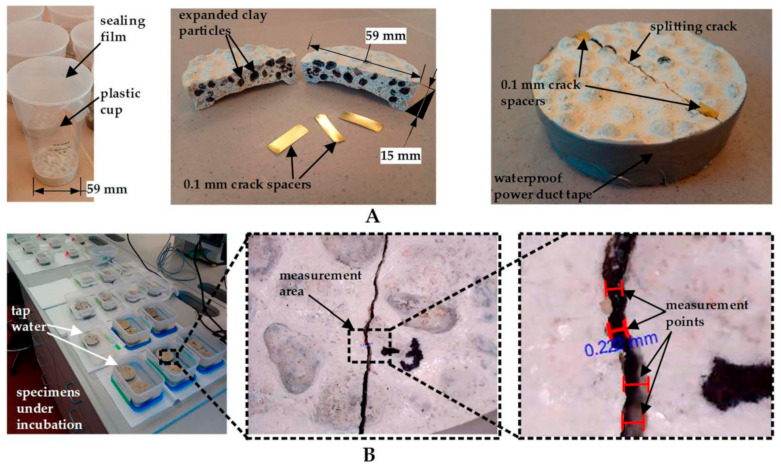
An illustrative representation of the sample preparation (**A**) and crack width measurement procedure (**B**).

**Figure 4 materials-14-02719-f004:**
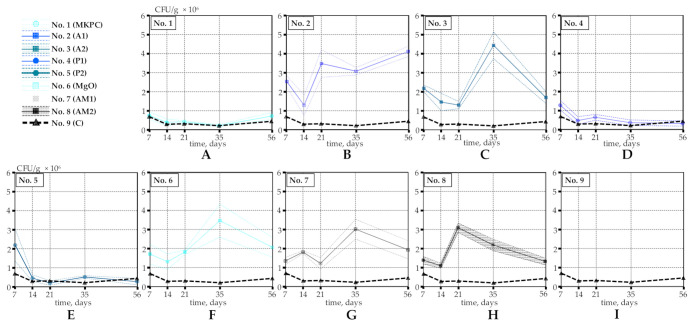
Bacterial viability in concrete specimens produced with expanded clay particles coated with: (**A**) potassium magnesium phosphate cement; (**B**,**C**) one and two layers of styrene–acrylate emulsion, respectively; (**D**,**E**) one and two layers of waterborne, latex and acrylic-composite paint, respectively; (**F**) magnesium oxide and magnesium hydroxide; (**G**,**H**) styrene–acrylate and magnesium oxide solution, prepared with a ratio 10:1 or 30:1, respectively; (**I**) uncoated control specimens. Solid lines in the graphs represent the average values, whereas shaded areas depict the 95% confidence intervals.

**Figure 5 materials-14-02719-f005:**
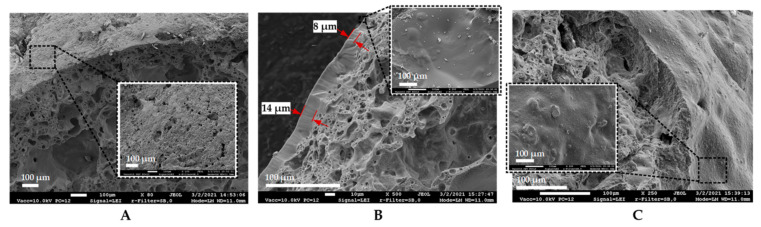
Scanning electron microscopy images of coated bacteria-immobilized expanded clay particles used in this study: (**A**) control sample with no coating and a vast number of open pores at the surface, (**B**) styrene–acrylate and (**C**) MgO-based coated samples with blocked pores. The variation of the thickness of a styrene–acrylate protective layer is indicated in (**B**).

**Figure 6 materials-14-02719-f006:**
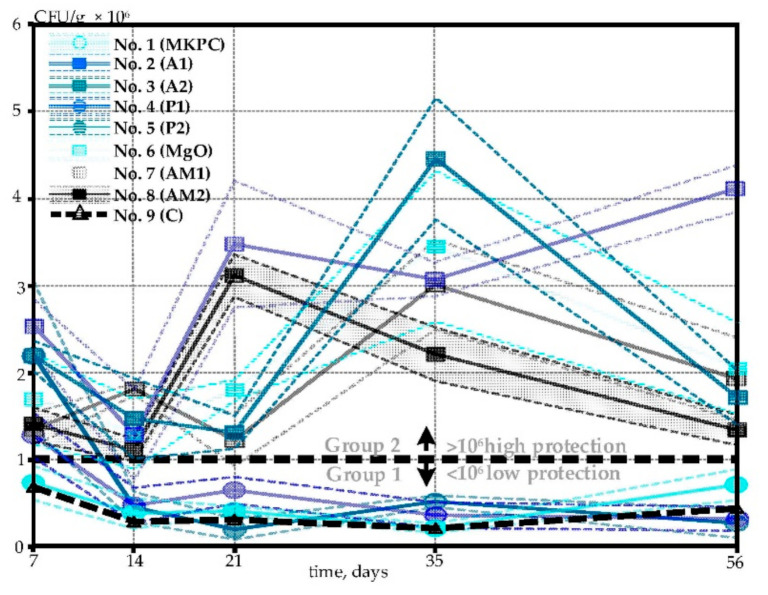
Comparison of the bacterial viability results in concrete specimens produced of differently coated bacteria-embedded expanded clay particles. Solid lines indicate the average values, whereas shaded areas refer to the 95% confidence intervals.

**Figure 7 materials-14-02719-f007:**
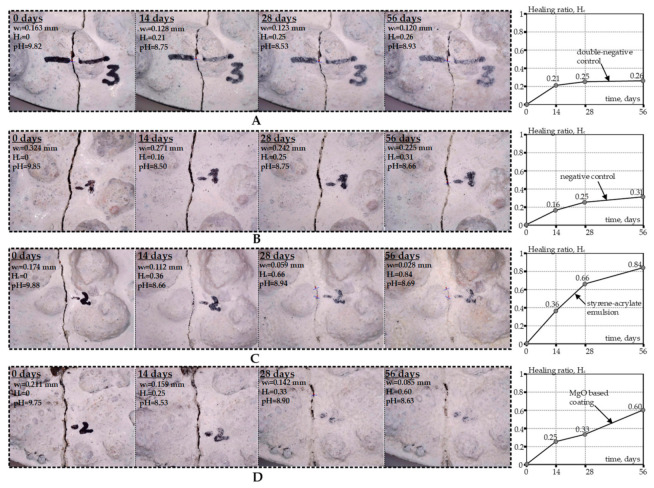
Optical measurement of crack healing in concrete specimens with coated expanded clay particles: (**A**) double-negative control with no coating and bacteria; (**B**) negative control specimen with embedded bacteria but no coating; (**C**) one layer of styrene–acrylate emulsion; (**D**) MgO-based coating. Here w_i_ is the initial crack width; w_t_ is the crack width after healing; H_c_ is the healing ratio, calculated H_c_ = (w_i_ − w_t_)/w_i_.

**Table 1 materials-14-02719-t001:** Coatings for the expanded clay used in this study.

No.	Coating	Composition
1	MKPC	Potassium magnesium phosphate cement
2	A1	One layer of styrene–acrylate emulsion
3	A2	Two layers of styrene–acrylate emulsion
4	P1	One layer of waterborne, latex, acrylic-composite paint
5	P2	Two layers of waterborne, latex, acrylic-composite paint
6	MgO	Magnesium oxide and magnesium hydroxide
7	AM1	Styrene–acrylate–magnesium oxide solution
8	AM2	Styrene–acrylate–magnesium oxide solution
9	C	No coating
10	CC	No coating, no bacteria

**Table 2 materials-14-02719-t002:** Concrete mix proportions.

No.	Material	Kg/M^3^	G/Sample
1	Sand, 0/4 mm	855	30
2	Cement, CEM-I	463	16.2
3	Water	231.5	8.1
4	Expanded clay, 4/8 mm	280	9.8

## Data Availability

The data presented in this study are available on request from the corresponding author.

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
