# Peer review of "Prolonging Bacterial Viability in Biological Concrete: Coated Expanded Clay Particles"

_materials, 2021, doi:10.3390/ma14112719_

Round 1

Reviewer 1 Report

This paper focuses on the protective materials of the capsules containing bacteria for enhancing the self-healing capability of concrete. A lot of experiments have been conducted, and the experimental design is well organized. The argument is also clear.
However, some points seem to be inadequately described. For example, the paper states that it is “a long term viability analysis”. However, the most extended test period is 56 days. This period is much shorter than the lifetime of general concrete structures. This study was conducted on the specific bacterial species, Alkaliphilic Bacillus pseudofirmus DSM 8715. However, it has not been shown if the results obtained are unique for this bacterial species or more generalizable. There is a need to specify and clarify the scope of this study and the results obtained from it. In addition, there are several points to be modified below.

- Figure 2;
Please shows the scale. The SEM images of Figures 1 and 5 also need readable scales.

- 2.4;
Please describe the details of the specimens, e.g., shape, size, mixing procedures, curing conditions, splitting [22] and tightening methods, and so on.

- Figure 4 and the first paragraph of 3.1;
I understand that the study focuses on a long-term protective capability, and only later than 7-day results are shown. However, the reference [16] is an unpublished paper, and readers cannot get any information. A detailed description within the starting 7-day changes is not needed, but the initial condition (0-day data) is mandatory in Figure 4.

- Figure 4;
Are the initial values of all series the same? In the cases of No. 2, 3, 6, 7, and 8, the numbers of live bacteria increased at 21 and/or 35 days. Please give discussions for this interesting trend.
The legend of No. 9 is the dashed line, but the solid line is shown. Please modify it. The same problem can be found in also Figure 6.

- 3.2;
Please define “Healing ratio Hc”, although this definition is shown in the reference [27].
There is no graph showing Hc results. A diagram with the bacterial viability (Figure 4) vs. Hc might be helpful to understand the efficiency of the protective materials.

- Figure 7;
Shown photos are too small. Please make them readable.

Author Response

Referee #1

This paper focuses on the protective materials of the capsules containing bacteria for enhancing the self-healing capability of concrete. A lot of experiments have been conducted, and the experimental design is well organized. The argument is also clear.

We would like to thank the Referee for reviewing our manuscript and for his/her interest in the topic.

However, some points seem to be inadequately described.

  • For example, the paper states that it is “a long term viability analysis”. However, the most extended test period is 56 days. This period is much shorter than the lifetime of general concrete structures.

We thank the Referee for remarks. We agree that the time period of 56 days used in the study is incomparable to the service life of the concrete structures. We adopted terms “short” and “long” from concrete creep and shrinkage tests [1]. Here any structural test that lasts more than 28 days is classified as a long-term test.

  1. Ojdrovic, R. P., & Zarghamee, M. S. (1996). Concrete creep and shrinkage prediction from short-term tests. Materials Journal, 93(2), 169-177.

  • This study was conducted on the specific bacterial species, Alkaliphilic Bacillus pseudofirmus DSM 8715. However, it has not been shown if the results obtained are unique for this bacterial species or more generalizable. There is a need to specify and clarify the scope of this study and the results obtained from it.

We thank the Referee for the important comment. Following the recommendations, we have clarified the scope of the study, as well as the results section:

  • A sentence clarifying the scope of the study was added to the last paragraph of the first section: To evaluate the performance of coating materials, B. pseudofirmus bacterial strain comonly used as producer of a healing agent was selected [8, 19, 20]
  • The discussion of the results in the last paragraph of Section 3.1 was modified: However, more data including the impact of different bacterial strains, concrete mix, curing conditions as well as sample age and size is required to give final conclusions.
  • Two clarifying sentences were added in the Conclusions:

In present study a detailed investigation on viability of B. pesudofirmus bacteria in concrete specimens produced with differently coated EC particles has been carried out.

Undoubtedly, future research is needed to assess the protective capabilities of styrene-acrylate and MgO-based coatings for other bacterial strains suitable for biological concrete production.

In addition, there are several points to be modified below.

  • Figure 2. Please show the scale. The SEM images of Figures 1 and 5 also need readable scales.

The recommended modifications were acknowledged. The scale was shown in Figure 2. The scales in Figures 1 and 5 were also updated.

  • 4. Please describe the details of the specimens, e.g., shape, size, mixing procedures, curing conditions, splitting [22] and tightening methods, and so on.

As suggested, we added additional Figure 3A that summarizes the details of the specimen preparation, curing conditions, dimensions and tightening methods. 

  • Figure 4 and the first paragraph of 3.1. I understand that the study focuses on a long-term protective capability, and only later than 7-day results are shown. However, the reference [16] is an unpublished paper, and readers cannot get any information. A detailed description within the starting 7-day changes is not needed, but the initial condition (0-day data) is mandatory in Figure 4.

We thank the Referee for raising this issue. According to this comment, we have described the initial number of bacterial spores in the dry expanded clay particles:

  • Last sentence of Section 2.2: 2·107 bacterial spores were immobilized into one gram of dry EC on average.
  • Second paragraph of Section 3.1: The initial number of bacterial spores was equal in all specimens (2.2·107 CFU/g, see Section 2.2). Notably, a 10-fold or in some cases even 100-fold decrease in a number of viable spores was observed in the first 7 days of concrete curing (Fig. 4). Similar results were reported in [14]. After the initial 7-day period, the number of viable bacterial spores stabilized.

  • Figure 4. Are the initial values of all series the same? In the cases of No. 2, 3, 6, 7, and 8, the numbers of live bacteria increased at 21 and/or 35 days. Please give discussions for this interesting trend.

As suggested, we have discussed this variation of viable bacterial spores in the last paragraph of Section 3.1: In fact, the nature of viability testing and sample preparation (see Sections 2.4 and 2.6) makes the data dependent on the amount of healing agent that is collected in 6.53g of concrete sample. As such, the larger amount of collected EC particles may even result in a growing number of CFU over time, as may be noticed from Figures 4 and 6.

  • The legend of No. 9 is the dashed line, but the solid line is shown. Please modify it. The same problem can be found in also Figure 6.

The recommended modifications were acknowledged.

  • 2. Please define “Healing ratio Hc”, although this definition is shown in the reference [27].There is no graph showing Hc results. A diagram with the bacterial viability (Figure 4) vs. Hc might be helpful to understand the efficiency of the protective materials.

We thank the Referee for remarks. As suggested, we have added four diagrams in Figure 7 showing variation of healing ratio Hc over time. We also defined the healing ratio in the caption of Figure 7.

  • Figure 7. Shown photos are too small. Please make them readable.

Figure 7 was modified accordingly.

Reviewer 2 Report

  In this work, a coated EC has been investigated for its potential use as a bacterial carrier in biological concrete. Eight different materials for coatings were selected considering the cost, workability and accessibility. The authors got the conclusion that healing efficiency in biological concrete specimens is strongly related to viable bacteria present in the healing agent, and additional coating of EC particles improves long-term bacterial viability. It is an interesting work. I have several comments before I can suggest publishing. -The authors should give photos of samples before test. -The authors should explain the principle of bacterial action. -The manuscript should have more quantitative analysis and charts to illustrate the advantages of the proposed approach. -Some recent published papers are suggested. CONSTRUCTION AND BUILDING MATERIALS,vol 244,article No.118048,2020.

Author Response

Referee #2

In this work, a coated EC has been investigated for its potential use as a bacterial carrier in biological concrete. Eight different materials for coatings were selected considering the cost, workability and accessibility. The authors got the conclusion that healing efficiency in biological concrete specimens is strongly related to viable bacteria present in the healing agent, and additional coating of EC particles improves long-term bacterial viability. It is an interesting work.

We would like to thank the Referee for reviewing our manuscript and for his/her positive evaluation of the study.

 I have several comments before I can suggest publishing.

  • -The authors should give photos of samples before test.

As suggested, we have included additional photos of samples in Figure 3A.

  • The authors should explain the principle of bacterial action.

Following the recommendations, we have added a sentence and a reference describing the principle of bacterial healing in the first paragraph of Introduction: Several biological pathways have been proposed for the precipitation of CaCO3, such as metabolic conversion of organic acids, ureolysis or denitrification [1]

  • The manuscript should have more quantitative analysis and charts to illustrate the advantages of the proposed approach.

We thank the Referee for remarks. As suggested, we have revised the manuscript emphasizing the advantages of the proposed approach. We have included chart in Figure 7 comparing the healing ratio for concrete samples with differently coated expanded clay particles over time. From the shown data it may be evident that significantly higher healing ratio was achieved for specimens containing styrene-acrylate and MgO based coatings. Moreover, we have extended the discussion on the protective capabilities of coated expanded clay particles (Second paragraph of Section 3.1). Here we have shown that the number of viable spores in the coated expanded clay particles may be almost 10-fold higher in comparison to the uncoated ones.

  • Some recent published papers are suggested. CONSTRUCTION AND BUILDING MATERIALS,vol 244,article No.118048,2020.

We thank the Referee for the suggested paper. However we found that the suggested paper is out of the scope of the present study.
